

**Investigation of the functional relationship between antecedent rainfall and the probability**
**of debris flow occurrence in Jiangjia Gully, China**
Shaojie Zhang[1], Hongjuan Yang[1], Kaiheng Hu[1], Juan Ma[2], Dunlong Liu[3]
1. Key Laboratory of Mountain Hazards and Earth Surface Process, Institute of Mountain Hazards and Environment,
Chinese Academy of Sciences, Chengdu 610041, China
2. China Institute for Geo-Environment Monitoring, Beijing, 100081
3. College of Software Engineering, Chengdu University of Information and Technology, Chengdu, 610225, China
Correspondence to: Shaojie Zhang, E-mail: sj-zhang@imde.ac.cn; Kaiheng Hu, E-mail: khhu@imde.ac.cn
**Abstract**
A larger antecedent effective precipitation (AEP) indicates a higher probability of a debris flow
($P_{df}$) being triggered by subsequent rainfall. There are several scientific topics surrounding this
qualitative conclusion that can be raised, including what kinds of variation rules do they follow, and
whether there is a boundary limit. To answer these questions, Jiangjia Gully in Dongchuan, Yunnan
province, China, was chosen as the study area, and a numerical calculation, rainfall scenario
simulation, and Monte Carlo integration method were used to calculate the occurrence probability
of debris flow under different AEP conditions and derive the functional relationship between $P_{df}$ and
AEP. The relationship between $P_{df}$ and AEP can be quantified by a piecewise function, and $P_{df}$
reaches a maximum value of 18.96% after the AEP exceeds 110 mm, indicating that debris flow in
nature has an extremely small probability compared to the rainfall frequency. Data from 1094
rainfall events and 37 historical debris flow events were collected to verify the reasonability of the
functional relationship. The results indicate that the first two stages of the piecewise function are



highly correlated with the observation results. Our study confirms the correctness of the qualitative
description of the relationship between AEP and $P_{df}$, clarifies that debris flow is a small probability
event compared to rainfall frequency, and quantitatively reveals the evolution law of debris flow
occurrence probability with AEP, which can provide a clear reference for the early warning of debris
flows.
**Keywords:** Debris flow, antecedent effective rainfall, Dens-ID, Monte Carlo method
**1 Introduction**
The antecedent effective precipitation (AEP) is similar to a Trojan horse lurking inside a loose
soil mass, which can cooperate with subsequent rainfall at any time to trigger debris flow in a gully.
The AEP is equivalent to the preservation of precipitation in the soil mass before the triggering
rainfall process; it represents the saturation degree of the loose soil mass (Segoni et al., 2018a;
Leonarduzz and Molnar, 2020). Therefore, the soil moisture that has accumulated from antecedent
rainfall since the beginning of a rainfall season has a significant influence on how new storm rainfall
interacts with the loose soil mass within a gully (Fiorillo and Wilson, 2004; Long et al., 2020). If a
loose solid material is provided by shallow landslides or channel erosion, its shear strength is
decreased by an increase in AEP (Papa, et al., 2013; Senthilkumar et al., 2017; Liu et al., 2020), and
in the subsequent rainfall process, the supply rate of solid material resources can be significantly
enhanced (Wei et al., 2008; Bennett et al., 2014; Zhang et al., 2020). Additionally, increased AEP
and moisture content have been shown to enhance surface rainfall-induced runoff in a variety of
environments (Tisdall, 1951; Luk, 1985; Le Bissonnais et al., 1995; Castillo et al., 2003; Jones et
al., 2017; Hirschberg et al., 2021). Thus, AEP plays an important role in the formation of debris



flows (Hong et al., 2018).
The rainfall threshold represents the degree of difficulty of debris flow triggered by rainfall
(Marra et al., 2017). Investigations, such as the influence of AEP on the rainfall threshold, can be
helpful in examining the relationship between AEP and debris flow occurrence. Currently,
conclusions drawn from the analysis of the relationship between the AEP and rainfall threshold are
relatively consistent, and there is a negative correlation between the AEP and rainfall conditions
(such as daily rainfall) that trigger debris flows (Huang, 2013). AEP represents the degree of
saturation of the loose soil mass (Zhao et al., 2019a; Abraham et al., 2021), and integrating soil
moisture with rainfall thresholds has been proven effective in improving these thresholds (Segoni
et al., 2018a; Zhao et al., 2019b; Abraham et al., 2020), as the antecedent moisture content plays a
key role in the soil shear strength. Scholars have attempted to analyze the influence of antecedent
soil moisture on the rainfall threshold triggering debris flow (Cui et al., 2007; Hu et al., 2015).
Similar to the relationship between AEP and rainfall threshold, there is a negative correlation
between antecedent soil moisture and triggering rainfall conditions (Chen et al., 2017). The above
investigations on the AEP and antecedent soil moisture show that the AEP can significantly decrease
the rainfall conditions that trigger a debris flow, which in turn means that debris flow is more likely
to occur. Therefore, there is the following consensus in the field of debris flow: the greater the AEP,
the higher the probability ($P_{df}$) of subsequent rainfall triggering the debris flow (De Vita et al., 2000;
Bel et al., 2017). Therefore, discovering a specific function to describe this qualitative description
is helpful in further demonstrating the above consensus, revealing a certain evolutionary law of
debris flow with rainfall in nature. Long-term observational data may be used to achieve this purpose;
however, the number of debris flow gullies with long-term observational data worldwide is less than



10 (Hürlimann et al., 2019). Even at a field site, such as Jiangjia Gully, it has been difficult to provide
sufficient observational data to accomplish this goal for more than 60 years.

To quantify the evolution law of $P_{df}$ with AEP variation, a numerical model that can correlate

the rainfall parameters (I and D) with the debris flow density (Zhang et al., 2020; Long et al., 2020)
was denoted as the Dens-ID model and was used to construct the rainfall intensity-duration (ID)
threshold curve database for different AEP. The ID threshold curves with upper and lower bounds
can delineate the closed region in the ID coordinate system, which represents the set of all rainfall
conditions that can trigger debris flow at a certain AEP. Consequently, the probability of natural
rainfall falling into a closed region is equivalent to $P_{df}$, which can then be calculated based on Monte
Carlo integration. The next section introduces the basic information of study area including the
rainfall and debris flow event data collected from the study area. The third section addresses how to
establish the functional relationship between the AEP and $P_{df}$ using the Dens-ID and Monte Carlo
integration method. Section 4 and 5 discuss the results and state the conclusions of this study,
respectively.
**2 Study area**

The Jiangjia Gully (JJG), a primary tributary of the Xiaojiang River, is located in the

Dongchuan District of Kunming City, Yunnan Province, China (Fig.1). As shown in Fig.1, JJG has
a drainage area of 48.6 km² with elevations ranging from 1040–3260 m. In this gully, the relative
relief from the ridge to the valley reaches 500 m, and most of the slope gradient is greater than 25°.
Slopes within JJG are covered by abundant loose soil with a thickness of more than ten meters.
Shallow landslides are frequently triggered by intense rainfall processes in JJG, providing a large
amount of solid materials for debris flow (Yang et al., 2022). The Menqian and Duozhao gullies,



shown in Fig.1, are the two main tributaries of JJG, accounting for 64.7% of the entire drainage area.
The upstream areas of the two main tributaries are the initiation zones of the debris flows, and the
channels of the upstream tributaries are narrow and V-shaped (Zhang et al., 2020).

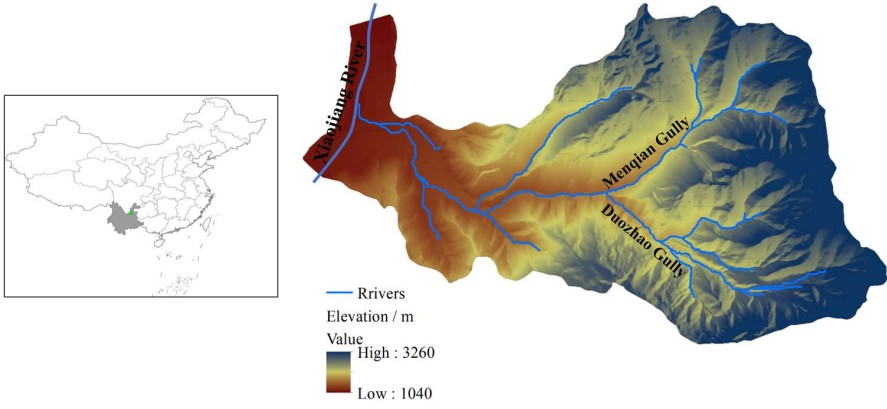


Fig.1 Location of JJG

Steep terrain provides a beneficial potential energy condition for transporting a large amount

of loose solid materials from JJG to Xiaojiang River. Consequently, debris flows in JJG can be easily
triggered by rainfall. Based on the collected rainfall data, high-intensity rainstorm or long-duration
rainfall processes can cause debris flow occurrence (Zhang et al., 2020). The solid material
necessary for a debris flow in a gully may be sourced from shallow landslides (Iverson et al., 1997;
Gabet and Mudd, 2006; Zhang et al., 2020; Long et al., 2020) or runoff-induced bed erosions (Berti
and Simoni, 2005; Coe et al., 2008; Tang et al., 2020; Bernard and Gregoretti, 2021). In JJG, the
solid material is sourced primarily from shallow landslides (Zhang et al., 2014; Liu et al., 2016;
Yang et al., 2022), which is consistent with the assumptions of Dens-ID (Zhang et al., 2020). Thus,
JJG is used as the study zone for deriving the function that describes the relationship between AEP
and $P_{df}$.

The JJG datasets for Dens-ID are terrain data, hydrological parameters, and soil mechanical



parameters. The DEM is the basal data for deriving other terrain data, including slope length,
gradient, and river channels; the spatial resolution of the DEM is 0.5 m, and a DEM with a grid size
of 10 m was generated using the resampling technology in ArcGIS. The hydrological parameters
are related to the soil types within JJG; the five key parameters are the saturated soil water content,
residual soil water content, the two parameters of soil water characteristic curve including $n$ and $m$,
and the infiltration rate of topsoil. The soil mechanical parameters are the soil cohesion force and
internal friction angle, which were obtained through direct shear tests on the soil samples. Detailed
data are available in Zhang et al. (2020) and Long et al. (2020).

Rainfall data for the rainy seasons between 2006 and 2020 were collected from the JJG

observation station, and it was necessary to identify each rainfall process from the long-term rainfall
sequences. Inter-event time (IET) was defined as the minimum time interval between two
consecutive rainfall pulses (Adams et al., 1986). IET has a strong influence on the rainfall event
starting and ending times (Bel et al., 2017), and Peres et al. (2018) identified that IET depends on
whether the mean daily potential evapotranspiration (MDPE) is larger than precipitation within the
IET. The long observation of evaporation within JJG showed that MDPE is about 4 mm;
precipitation during IET >0.5 mm is considered the end of a rainfall process. Under this standard,
1094 rainfall events and 37 debris flow events were identified during the sampling period. Detailed
rainfall data information can be found in "appendix 1-1094 rainfall and 37 debris flow data.xlsx".
The AEP listed in this appendix was considered the weighted sum of the rainfall periods before the
occurrence of debris flow (Long et al., 2020) and it can be calculated using Eq. 1.
$$AEP = \sum_{i=1}^{n} K^n R_i \qquad (1)$$



where AEP is the antecedent effective rainfall; $K$ is the attenuation coefficient, which is equal to
0.78 based on the field test in JJG (Zhang et al., 2020); and $n$ is the number of days preceding the
debris flow occurrence.
Based on the observed rainfall data, the 1094 AEPs were calculated using Eq. 1 and are listed
in Appendix 1. The AEP corresponding to each rainfall event varies from 0–88 mm. Taking this
variation range as a reference, the variation range of the AEP input in the Dens-ID model was set
between 10 and 130 mm. When the AEP was less than 90 mm, it was gradually increased by 5 mm;
after the AEP was larger than 90 mm, its increment was set to 10 mm. Dens-ID presets several AEP
values including 10, 15, 20, 25, 30, 35, 40, 45, 50, 55, 60, 65, 70, 75, 80, 85, 90, 100, 110, 120, and
130 mm. $P_{df}$ can be calculated under different AEP conditions. The preset AEP values exceeded the
observed maximum value of 88 mm because we wanted to observe whether $P_{df}$ tended to stabilize
and determine its boundary value
**3 Methods**
**3.1 Dens-ID**
Debris flow gullies, characterized by a solid source supply from landslides, are widely
distributed in southwest China (Zhang et al., 2014). For this type of debris flow gully, our previous
study proposed a numerical model (denoted as Dens-ID) based on the evolution law of fluid density
(Zhang et al., 2020; Long et al., 2020). Den-ID assumes the debris flow to be a water-soil mixture.
Based on the digital elevation model (DEM) of a gully, Den-ID, which uses a grid cell as a basic
mapping unit, can simulate the surface runoff and water diffusion in the vertical direction within the
soil mass.
$$-D(\theta)\frac{\partial \theta}{\partial z} + K(\theta) = I(t) \qquad (2)$$



where $\theta$ is the soil water content; $D(\theta) = K(\theta)/(d\theta/d\psi)$, which represents the soil water
diffusivity; $z$ is the soil depth, which is positive downwards along the soil depth as the topsoil is
taken as the origin point; $K(\theta)$ is the hydraulic conductivity; $I(t)$ is the rainfall intensity; and $\psi$ is
the soil matrix suction. When the rainfall intensity was less than the surface infiltration capacity, Eq.
2 was used to represent this physical process. The case of precipitation intensity exceeding the
infiltration capacity of topsoil means that the surface is saturated, and the excess precipitation from
the topsoil is typically converted into runoff; therefore, the pressure infiltration of each grid cell is
not considered. As the topsoil is saturated by rainfall, Eq. 2, which controls the infiltration border,
uses $\theta = \theta_s$, where $\theta_s$ is the saturated water content of a soil type within a debris flow gully.
$$\frac{\partial \theta}{\partial t} = \frac{\partial}{\partial z}\left[D(\theta)\frac{\partial \theta}{\partial z}\right] - \frac{\partial K(\theta)}{\partial \theta} \tag{3}$$

Eq. 3 is the Richard differential infiltration equation (Richards, 1931), which is used to describe the
water movement law along the vertical direction within the soil mass after precipitation infiltrates
the topsoil. Dens-ID uses the finite-difference method to solve Eqs. 2 and 3 and can provide the
runoff depth (denoted as $dw(i,t)$ ), soil water content, and soil matrix suction for each grid cell.
Dens-ID then calculates the runoff volume using runoff depth $dw(i, t)$ in Eq. 4.
$$V_w(t) = \sum_{t=1}^{T}\sum_{i=1}^{n} S_g * dw(i,t) \tag{4}$$

where $n$ represents the total number of grid cells that can generate runoff at time t and $V_w(t)$
represents the total volume of runoff within a gully at time $t$.

Taking hydrological parameters such as soil water content and soil matrix suction as inputs,

Dens-ID uses Eqs. 5 and 6 to estimate the supply volume of rainfall-induced loose solid materials
within a gully. Eq. 5 calculates safety factor $F_s$ of each grid cell as a function of the matrix suction
and soil moisture. $F_s > 1$ indicates that the grid cell is stable and cannot supply solid material to the



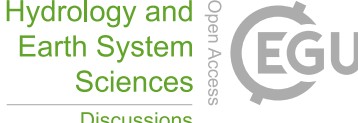

gully, whereas a grid with $F_s < 1$ can provide solid material in the form of a shallow landslide.
$$F_s = \frac{\tan \varphi}{\tan \beta} + \frac{c + \psi \, ta \ (\varphi^b)}{\gamma_t d_s \cos \beta \sin \beta} \qquad (5)$$
where $F_s$ represents the safety factor of each grid cell, $c$ is the soil cohesion force, $\varphi$ is the internal
friction angle, $\varphi^b$ is related to the matrix suction and is approximately equal to $\varphi$ as the low matrix
suction is small, $d_s$ is the soil depth, and $\psi$ is the matrix suction of the soil, a function of soil water
content, and can be described by the Van Genuchten model (Van Genuchten, 1980).

Using $d_s$ derived from Eq. 5 as input, Eq. 6 is used to estimate the total volume of solid

materials provided from all the instable grid cells in the gully from the beginning to the end of the
rainfall process.
$$V_s(t) = \sum_{t=1}^{T} \sum_{j=1}^{m} S_g * ds(j, t) \qquad (6)$$
where $m$ represents the number of grid cells that can provide solid material at time $t$ and $V_s(t)$ is the
total volume of solid material within a gully at time $t$. At time $t$, the density of the water-soil mixture
after full coupling between runoff and solid material can be calculated using Eq. 7.
$$\rho_{mix}(t) = \frac{\rho_w V_w(t) + \rho_s V_s(t)}{V_{mix}(t)} \qquad (7)$$
where $\rho_{mix}(t)$ is the density of the water-soil mixture, $\rho_w$ is the water density, $\rho_s$ is the density of
the soil particles, and $V_{mix}(t)$ is the volume of the water-soil mixture, which is the sum of $V_w(t)$
and $V_s(t)$. $V_w(t)$ and $V_s(t)$ are the key variables that can be derived using Eqs. 4 and 6.

Dens-ID presets the density of the water-soil mixture as $\rho_{mix}$. By simulating many rainfall

scenarios, including long durations with low-intensity rainfall and short durations with high-
intensity rainfall, Dens-ID can obtain adequate combinations of [$D_i$, $I_i$]. Using each [$D_i$, $I_i$] as input,
Dens-ID derives the density value via hydrology simulation and estimate the solid material and
runoff volumes. When the calculated density is equal to $\rho_{mix}$, the [$D_i$, $I_i$] combination is saved by





Dens-ID. After Dens-ID completes the trial calculations, all combination data of [$D_i$, $I_i$] that satisfy
the constraints of the preset density ($\rho_{mix}$) can be collected, forming a dataset. Each collected [$D_i$, $I_i$]
within the dataset corresponds to $\rho_{mix}$; therefore, Dens-ID can map rainfall parameters (D and I) and
debris flow density (Long et al., 2020). Dens-ID can derive the ID threshold curves by fitting the
selected [$D_i$, $I_i$] data; each ID curve corresponds to a debris flow density value (Zhang et al., 2020).
As the density of debris flow in JJG varies in a specific interval of 1.2–2.3g/cm$^3$ (Zhang et al., 2014;
Zhuang et al., 2015; Long et al., 2020), the threshold curve that corresponds to the boundary value
can form a closed area with the I- and D-axes in the ID coordinate system. The case of monitoring
or forecasting rainfall falling into this closed area in the I-D coordinate system indicates that the
rainfall condition may trigger debris flow. The verification results for JJG show that Dens-ID
effectively describes the mechanism and process of debris flow formation using shallow landslides
as a solid source supply, and its prediction accuracy is approximately 80.5%, which is 27.7% higher
than that of statistical models (Zhang et al., 2020). Such a high prediction accuracy can further
indicate that the closed area formed by the derived ID curves has a very reasonable location and
coverage in the ID coordinate system, providing extremely reliable analytical data in this study.

.

**3.2 Monte Carlo method for calculating the definite integral**

Because of the boundary of the debris-flow density in JJG (1.2–2.3g/cm$^3$), Dens-ID produces

the corresponding upper and lower boundary curves under a specific AEP condition. The two
boundary curves can be described using the power function
$$\begin{cases} f(D)_{up} = I_{up} = \alpha_1 D^{\beta_1} & D\epsilon[a_1, b_1] \\ f(D)_{low} = I_{low} = \alpha_2 D^{\beta_2} & D\epsilon[a_2, b_2] \end{cases} \quad (8)$$



These two threshold curves can form a closed warning area in the ID coordinate system,
denoted as $W_{ID}$. The independent variable (D) and dependent variable (I) in Eq. 8 also form a closed
rectangular region in the ID coordinate system, denoted as $R_{ID}$. In the ID coordinate system, the
coverage of $R_{ID}$ is larger than that of $W_{ID}$, as will be shown in detail in Section 4.1. Within $R_{ID}$, if
certain rainfall processes are located in $W_{ID}$, this rainfall condition can trigger debris flow. As long
as the probability of rainfall falling into the range of $W_{ID}$ under random conditions can be
determined, the occurrence probability of debris flow can be estimated for a specific AEP. Many
physical phenomena are stochastic in nature and governed by stochastic partial differential equations
with nondeterministic initial/boundary conditions or integral equations (Yan and Hong, 2014).
Albert (1956) proposed the Monte Carlo method for solving integral equations. This method was
subsequently used to estimate the peak flow and volume of debris flow (Donovan and Santi, 2017;
Paola et al., 2017), entrainment of the underlying bed sediment (Han et al., 2015), and risk
assessment (Calvo and Savi, 2009; Li et al., 2021). Based on the Monte Carlo principle (Peres and
Cancelliere, 2014), the probability of the rainfall condition within the $R_{ID}$ range falling into the $W_{ID}$
range can be determined using $W_{ID}/R_{ID}$. The physical meaning of the Monte Carlo solving definite
integral is the estimation of the area enclosed by the function curve and horizontal axis. Therefore,
the area of $W_{ID}$ can be calculated by the difference in the definite integral formula of the two
equations in Eq. 7.

$$W_{ID} = S_{up} - S_{low} = \int_{a1}^{b1} f(D)_{up} dD - \int_{a2}^{b2} f(D)_{low} dD \tag{9}$$

where $S_{up}$ and $S_{low}$ represent the area enclosed by the two threshold curves and the horizontal axis,
respectively, and a₁, b₁, a₂, and b₂ are the boundary values of D in the two curves. For the upper
boundary line (or lower boundary), if the probability distribution function of D between [a1, b1] is





$p$(D), Eq. 9 can be derived by substituting $p$(D) into Eq. 8, which is used to calculate $S_{up}$ and $S_{low}$.
$$\begin{cases} S_{up} = \int_{a_1}^{b_1} f(D)_{up} dD = \int_{a_1}^{b_1} \frac{f(D)_{up}}{p(D)} p(D) dD \approx \frac{1}{n} \sum_{k=1}^{n} \frac{f(D_i)_{up}}{p(D_i)} \\ S_{low} = \int_{a_2}^{b_2} f(D)_{low} dD = \int_{a_2}^{b_2} \frac{f(D)_{low}}{p(D)} p(D) dD \approx \frac{1}{n} \sum_{k=1}^{n} \frac{f(D_i)_{low}}{p(D_i)} \end{cases} \quad (10)$$

$$W_{ID} = \frac{1}{n} \sum_{k=1}^{n} \frac{f(D_i)_{up}}{p(D_i)} - \frac{1}{n} \sum_{k=1}^{n} \frac{f(D_i)_{low}}{p(D_i)} \quad (11)$$

where $n$ represents the number of random samples drawn from the variation range of D and $p(D_i)$
is the probability density distribution function of D in the interval [$a_1$,$b_1$] or [$a_2$,$b_2$]. The key to
solving Eq. 10 is sampling from $p(D)$. The following steps were used to explain how samples were
taken using $p(D_i)$.
Step 1: Based on the probability density distribution function $p$(D), the cumulative probability
distribution function can be derived by $cdf$(D)=$\int_{-\infty}^{b} f(D) dD$;
Step 2: Assume that $U^{(i)}$ obeys a uniform distribution within [0,1], which can be randomly collected
from this interval and denoted as $U^{(i)} \sim$U(0,1).
Step 3: Substitute $U^{(i)}$ into the inverse function of the cumulative probability distribution $cdf$(D) to
obtain random sample $D^{(i)}$, denoted by $D^{(i)} = cdf^{-1}(U^{(i)})$. Then, a dataset composed of $n$ data
points of $D^{(i)}$ was obtained.
Step 4: $W_{ID}$ can be calculated by substituting $n$ data points of $D^{(i)}$ into Eq. 10, and the $P_{df}$ ($P_{df} =$
$\frac{R_{ID}}{W_{ID}}$) corresponding to a specific AEP is determined. $P_{df}$ represents the probability that the
subsequent precipitation process may trigger debris flow for a certain AEP. Thus, the influence of
the AEP on the occurrence probability of debris flows can be quantified.
**3.3 Correlation analysis between numerical and observation results**

The relationship between the AEP-$P_{df}$ fit through the observational data was used as a reference

standard, and the correlation analysis method was used to verify the function of the AEP-$P_{df}$ derived





by Dens-ID. Correlation analysis was used to study the degree of linear correlation between
variables, which is represented by correlation coefficient $r$:
$$r = \frac{\sum_{i=1}^{n}(x_i-\bar{x})(y_i-\bar{y})}{\sqrt{\sum_{i=1}^{n}(x_i-\bar{x})^2 \sum_{i=1}^{n}(y_i-\bar{y})^2}} \quad\quad\quad (12)$$
where $x$ represents the $P_{df}$ derived from the observed data, $y$ represents the $P_{df}$ derived from Dens-
ID, $\bar{x}$ and $\bar{y}$ represent the averages, $r$ represents the correlation coefficient, and $n$ represents the
number of samples. $|r|\geq 0.8$ can be regarded as a high correlation between two variables; $0.5\leq|r|<0.8$
represents a moderate correlation; $0.3\leq|r|<0.5$ represents a low correlation; and $|r|<0.3$ indicates the
degree of correlation between the two variables is weak and can be regarded as uncorrelated.
**4 Results and discussion**
**4.1 ID threshold curves and warning zone closed by the derived curves**
The ID threshold curves corresponding to the different AEPs derived from Dens-ID are listed
in Table 1. Each AEP corresponded to the upper and lower boundary lines of the ID threshold, and
these two boundary lines corresponded to different debris flow density values. In Table 1, when
AEP≤15 mm, the maximum density corresponding to the ID threshold curve cannot reach 2.2, which
are equal to 1.8 and 2.0 when AEP=10 and 15 mm. This is because a lower AEP makes the supply
rate of solid resources in JJG far less than the runoff rate during rainfall (Long et al., 2020). At this
time, Dens-ID determines that it is easier to form a low-density water-soil mixture in JJG.

Table 1 ID threshold curve database under different AEP

| AEP (mm) | ID threshold curve function for JJG | |
| --- | --- | --- |
| | 1.2 g/cm$^3$ | 2.2 g/cm$^3$ |
| 10 | $I_{1.2} = 19.85D^{-0.54}$ D∈[1, 269] ($R^2 = 0.991$) | **$I_{1.8}$**=15.85D$^{-0.48}$ D∈[1, 263] ($R^2$=0.990) |
| 15 | $I_{1.2}$= 21.69D$^{-0.55}$ D∈[1, 236] ($R^2$=0.993) | **$I_{2.0}$**=16.10D$^{-0.50}$ D∈[1, 229] ($R^2$=0.995) |
| 20 | $I_{1.2}$=23.22D$^{-0.58}$ D∈[1, 203] ($R^2$=0.996) | $I_{2.2}$=17.20D$^{-0.53}$ D∈[1, 192] ($R^2$=0.995) |
| 25 | $I_{1.2}$=24.47D$^{-0.60}$ D∈[1, 171] ($R^2$=0.997) | $I_{2.2}$=16.92D$^{-0.53}$ D∈[1, 160] ($R^2$=0.998) |





| 30 | $I_{1.2}=26.24D^{-0.64}$ $D\in[1, 143]$ $(R^2=0.996)$ | $I_{2.2}= 18.09D^{-0.57}$ $D\in[1, 132]$ $(R^2=0.995)$ |
|---|---|---|
| 35 | $I_{1.2}=35.47D^{-0.65}$ $D\in[1, 123]$ $(R^2=0.958)$ | $I_{2.2}= 19.55D^{-0.58}$ $D\in[1, 112]$ $(R^2=0.985)$ |
| 40 | $I_{1.2}= 40.59D^{-0.78}$ $D\in[1, 103]$ $(R^2=0.966)$ | $I_{2.2}= 22.15D^{-0.64}$ $D\in[1, 92]$ $(R^2=0.984)$ |
| 45 | $I_{1.2}= 41.12D^{-0.78}$ $D\in[1, 83]$ $(R^2=0.932)$ | $I_{2.2}= 23.19D^{-0.69}$ $D\in[1, 72]$ $(R^2=0.981)$ |
| 50 | $I_{1.2}=41.26D^{-0.86}$ $D\in[1, 65]$ $(R^2=0.981)$ | $I_{2.2}= 23.50D^{-0.74}$ $D\in[1, 55]$ $(R^2=0.980)$ |
| 55 | $I_{1.2}=38.63D^{-0.88}$ $D\in[1, 53]$ $(R^2=0.950)$ | $I_{2.2}= 23.31D^{-0.70}$ $D\in[1, 42]$ $(R^2=0.932)$ |
| 60 | $I_{1.2}= 31.49D^{-0.92}$ $D\in[1, 40]$ $(R^2=0.992)$ | $I_{2.2}= 20.73D^{-0.86}$ $D\in[1, 30]$ $(R^2=0.977)$ |
| 65 | $I_{1.2}= 29.14D^{-0.95}$ $D\in[1, 32]$ $(R^2=0.957)$ | $I_{2.2}= 18.10D^{-0.91}$ $D\in[1, 22]$ $(R^2=0.893)$ |
| 70 | $I_{1.2}= 23.05D^{-0.96}$ $D\in[1, 25]$ $(R^2=0.998)$ | $I_{2.2}= 13.04D^{-0.93}$ $D\in[1, 15]$ $(R^2=0.995)$ |
| 75 | $I_{1.2}= 21.13D^{-0.97}$ $D\in[1, 22]$ $(R^2=0.994)$ | $I_{2.2}=10.90D^{-0.95}$ $D\in[1, 12]$ $(R^2=0.995)$ |
| 80 | $I_{1.2}= 18.72D^{-0.98}$ $D\in[1, 20]$ $(R^2=0.997)$ | $I_{2.2}= 9.96D^{-0.95}$ $D\in[1, 11]$ $(R^2=0.999)$ |
| 85 | $I_{1.2}= 18.47D^{-0.99}$ $D\in[1, 18]$ $(R^2=0.999)$ | $I_{2.2}= 8.17D^{-0.95}$ $D\in[1, 9]$ $(R^2=0.999)$ |
| 90 | $I_{1.2}= 16.99D^{-0.98}$ $D\in[1, 18]$ $(R^2=0.999)$ | $I_{2.2}=6.81D^{-0.95}$ $D\in[1, 7]$ $(R^2=0.994)$ |
| 100 | $I_{1.2}= 16.90D^{-0.98}$ $D\in[1, 18]$ $(R^2=0.999)$ | $I_{2.2}=6.81D^{-0.95}$ $D\in[1, 7]$ $(R^2=0.994)$ |
| 110 | $I_{1.2}= 16.87D^{-0.98}$ $D\in[1, 16]$ $(R^2=0.999)$ | $I_{2.2}=6.76D^{-0.95}$ $D\in[1, 7]$ $(R^2=0.997)$ |
| 120 | $I_{1.2}= 16.87D^{-0.98}$ $D\in[1, 16]$ $(R^2=0.999)$ | $I_{2.2}=6.76D^{-0.95}$ $D\in[1, 7]$ $(R^2=0.997)$ |
| 130 | $I_{1.2}= 16.87D^{-0.98}$ $D\in[1, 16]$ $(R^2=0.999)$ | $I_{2.2}=6.76D^{-0.95}$ $D\in[1, 7]$ $(R^2=0.997)$ |

When AEP < 10 mm, Dens-ID cannot derive the threshold curve corresponding to even the
minimum density value of 1.2 g/cm³, which indicates that the subsequent rainfall can hardly trigger
debris flow JJG. Table 1 also shows that when the AEP reaches 110 mm, α and β in the threshold
curve become constant and no longer change with AEP. An AEP ranging from 10 to 110 mm can
affect the debris flow formation in JJG. After the AEP in Table 1 exceeded 90 mm, the effect of AEP
on the ID threshold curve was not significant.





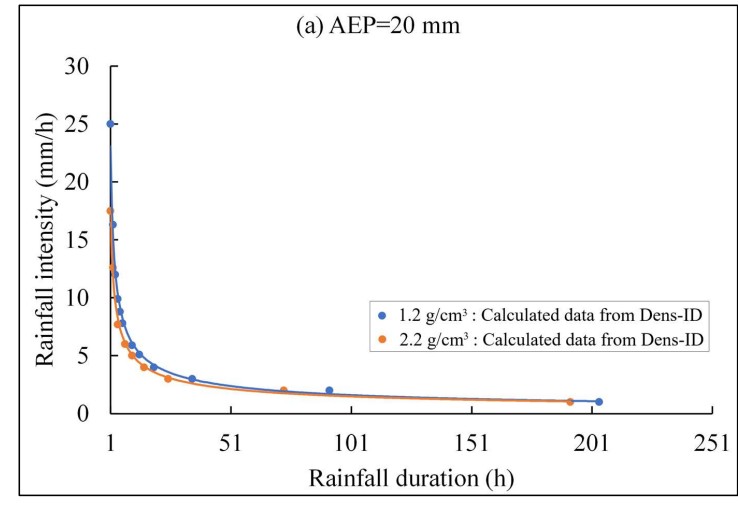





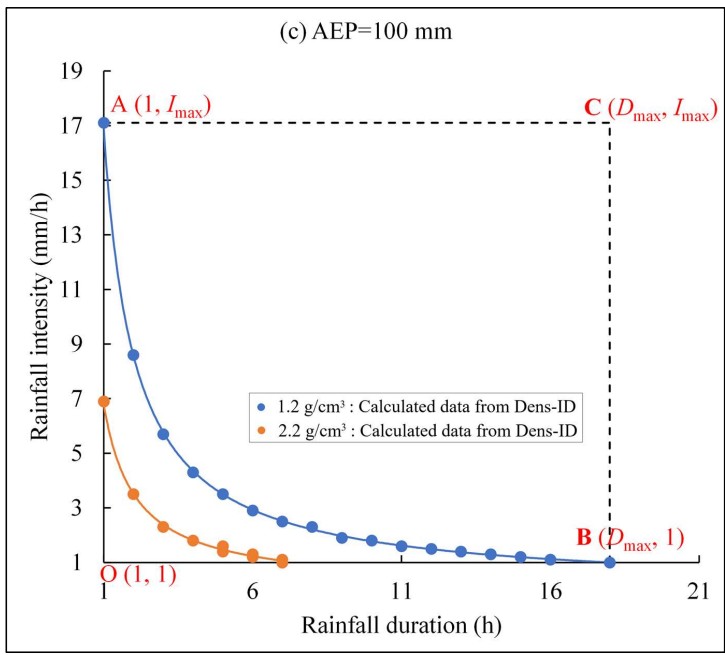

Fig.2 ID threshold curves derived by Dens-ID (the blue line corresponds to 1.2 g/cm$^3$, and the

orange line corresponds to 2.2 g/cm$^3$ in the figures)

There are two ID threshold curves in each subplot of Fig. 2, which correspond to 1.2 g/cm$^3$ and

2.2 g/cm$^3$, respectively. Because the debris flow density in JJG varies within a certain range from

1.2–2.3 g/cm$^3$, the two ID threshold curves shown in each subplot can be regarded as the upper and

lower boundary lines for determining the occurrence of debris flow (Zhang et al., 2020). As shown

in Fig.2c, the two derived curves, together with the I- and D-axes, form a closed area in the ID

coordinate system; this area is denoted as $W_{ID}$. If the monitored rainfall, represented by the

combination of I and D, can enter $W_{ID}$, rainfall may trigger debris flows. As shown in each subplot,

the threshold curve can be represented by power function $I=\alpha D^{\beta}$. The variation intervals of the

independent (D) and dependent (I) variables of the power function are [1, $D_{max}$] and [1, $I_{max}$],

respectively, where $D_{max}$ represents the rainfall duration required to trigger debris flow when I= 1





mm/h, and $I_{max}$ represents the rainfall intensity required for debris flow formation for D=1 h. As
shown in Fig.2c, independent variable D and dependent variable I can form a larger rectangular area
(AOBC) in the ID plane than $W_{ID}$, which is denoted as $R_{ID}$. The coverage area of $R_{ID}$ is much larger
than that of $W_{ID}$, indicating that the proportion of rainfall conditions that can trigger debris flows is
low. Therefore, even for AEP=100 mm, the occurrence probability of debris flows remains low. As
shown in each subplot, each AEP corresponds to a different $W_{ID}$ and $R_{ID}$, which provides basic data
for the quantitative evaluation of the effect of different AEPs on the occurrence probability of debris
flows.
**4.2 Occurrence probability of debris flow under different AEP**

Based on the Monte Carlo method of calculating the definite integral, it is necessary to explore

the probability density function of rainfall duration (D) to calculate the occurrence probability of
debris flow under different AEP conditions. For the 1094 rainfall events listed in Appendix 1, we
found that the probability distribution of rainfall duration D in JJG can be described by a power
function (Fig. 3). As shown in Fig.3, the number of samples with D<1 accounted for 37.7%, 1<D<3
for 23.5%, 3<D<5 for 14.7%, and 5<D<10 for 16.9%; the number of rainfall events with D
exceeding 10 h accounted for only 6.7%.



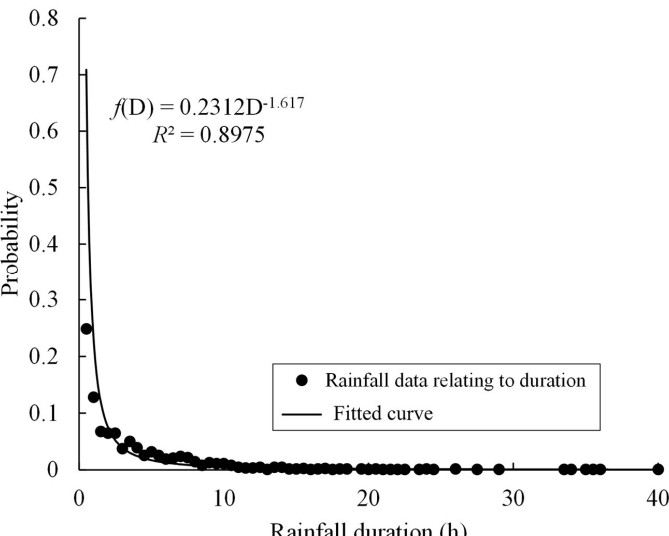


Fig. 3 Probability density function of $f$(D)

Based on the probability density distribution function $f$(D)=0.2312D$^{-1.617}$, the cumulative
probability function $cdf$(D) can be obtained through integration. In $cdf$(D), denoted as Eq. 11, the
integration constant $C$ needs to be determined.

$$cdf(D) = \int_{-\infty}^{D} f(D)dD = -0.3747 * D^{-0.617} + C \qquad (11)$$

The interval of 0–40 h was evenly divided into 56 statistical intervals (the second column in
Appendix 2, titled "appendix 2-f(D)and CFD(D). xlsx"), and each statistical interval was separated
by 0.5 h. The proportion of the sample size in each interval among the 1094 samples can be
calculated (second column in Appendix 2), and the cumulative proportion that increases with D is
obtained (third column in Appendix 2). The data in the first and third columns of Appendix 2 are
substituted into Eq. 11 to calculate $C$. $C$ increases with D but gradually stabilizes at approximately
1.04 (the fifth column in Appendix 2), and $C$ is set to 1.04.
Based on the process of calculating $P_{df}$ under different AEP conditions in Section 3.4, the $P_{df}$
corresponding to each AEP in Table 1 was obtained, and the function $P_{df} = f(AEP)$ for describing



their relationship was fitted using the AEP and $P_{df}$ data.
$$\begin{cases} P_{df} = 0 & 0 < AEP < 10 \\ P_{df} = 0.34e^{0.046AEP} & 10 \leq AEP < 85 \\ P_{df} = 0.1AEP + 7.6 & 85 \leq AEP < 110 \\ P_{df} = 18.96 & 110 \leq AEP \geq 130 \end{cases} \quad (12)$$

As shown in Eq.12, $P_{df} = f(AEP)$ is a piecewise function. The evolution of $P_{df}$ with AEP
variation can be divided into four stages (Fig. 4). Two key issues must be stated before discussing
these four stages in depth: (1) Based on the calculation results of the Dens-ID model, an upper limit
volume of the rainfall-induced solid material supply is derived in JJG, which is the basic condition
for determining the scale of debris flow in JJG (Zhang et al., 2020). (2) Based on the principle of
water balance, AEP is defined as the rainfall that is preserved in the soil before the triggering rainfall
process (Kohler and Linsley, 1951); field observations in JJG show that the AEP is positively
correlated with the soil water content (Cui et al., 2007), and the field observations of the Liudaogou
catchment in the northern Loess Plateau of China have the same result (Zhu and Shao, 2008);
therefore, the AEP is typically used to estimate soil water content (Crozier, 1986; Chen et al., 2018;
Zhao et al., 2019b). The water soil content before the triggering rainfall process can be characterized
by AEP (Thomas et al., 2019; Schoener and Stone, 2020).





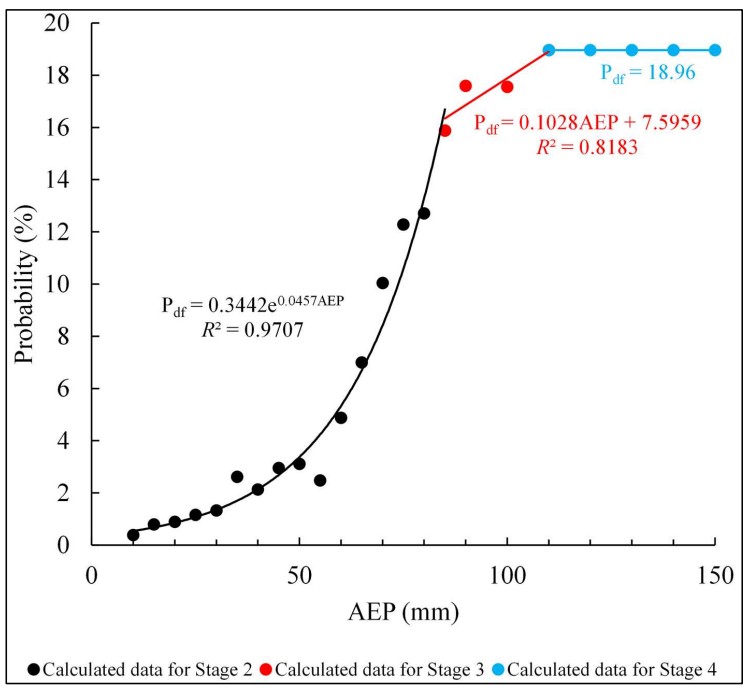


Fig.4 Relationship of $P_{df}$ and AEP derived from Dens-ID (the black line represents the fitted curve

of Stage 2, the red line represents the fitted curve of Stage 3, the blue lien represents the fitted

curve of Stage 4)

Stage 1: The probability of debris flow occurrence in JJG is equal to 0 when the AEP is < 10

mm. Dens-ID estimates the solid material volume by simulating rainfall-induced shallow landslides.

According to Eq. 4, the key hydrological process that triggers shallow landslides is the continuous

increase in soil water content caused by rainfall infiltration. The increase in soil moisture content

reduces soil matrix suction and eventually contributes to shallow landslides. The soil water content

of the loose soil mass in JJG was low when the AEP was < 10 mm (Long et al., 2020), and a long

duration of rainfall infiltration was needed to increase the soil water content. However, based on the

infiltration border of Dens-ID (Eq. 1), limited by the infiltration capacity of the topsoil in JJG, the

portion of precipitation that exceeds the infiltration capacity is be converted into runoff; therefore,



when the water content of the soil layer in JJG is low, the surface runoff has already been generated.
Accordingly, the runoff generation rate can be much higher than the supply rate of solid material in
the Dens-ID simulation. In this hydrological scenario, Dens-ID determines that even a soil-water
mixture with a density of 1.2 g/cm$^3$ is difficult to generate in JJG; thus, the probability of debris
flow is 0.

Stage 2: The relationship between $P_{df} \sim AEP$ can be described by an exponential function,

indicating that the probability of debris flow occurrence is enhanced by gradually increasing AEP.
This trend obeys the following function: $P_{df} = 0.3442e^{0.0457AEP}$, which can be further divided into
two subprocesses using AEP = 50 mm as the demarcation point, where the slope of the curve
changes significantly. Stage 2-1: When 10 mm≤AEP≤50 mm, the soil water content increased
significantly compared to AEP < 10 mm, but a necessary infiltration time to increase it to the critical
state for triggering shallow landslides is still required. Therefore, limited by the supply rate of the
solid material, the rate of increase of $P_{df}$ was relatively low, and the maximum $P_{df}$ was 3.11%. Stage
2-2: When 50 mm<AEP≤85 mm, the soil water content is relatively large compared to Stage 1; the
solid material from shallow landslides can be immediately ready without a long rainfall infiltration
duration, and a large soil water content of topsoil is beneficial to the rapid generation of runoff
(Jones et al., 2017; Hirschberg et al., 2021). When there is a sufficient supply of provenance and
runoff, the probability of debris flow occurrence in this subprocess is significantly enhanced by the
increasing AEP.

Stage 3: After the AEP exceeded 85 mm, the rate of increase of $P_{df}$ decreased, exhibiting a

moderate linear increasing trend with AEP. Because of the very high soil water content, most of the
loose soil layer in JJG is close to the saturated state (Long et al., 2020). Then, the total volume of





solid material reaches the maximum level, and the increased AEP can hardly contribute to the runoff
generation rate. Consequently, the increasing trend of $P_{df}$ slows compared with that in Stage 2-2.
Stage 4 (AEP≥ 110 mm): According to the ID threshold curves in Table 1, the two key
parameters $\alpha$ and $\beta$ of the threshold curve at this stage are already in a constant state, which means
that there is no longer any change in $R_{ID}$ and $W_{ID}$ in Fig. 2c. Therefore, the $P_{df}$ no longer changed
with increasing AEP and remained unchanged at 18.96%.
**4.3 Correlation analysis of the two curves derived from Dens-ID and observation data**
The AEP in Appendix 1 varied from 0–87.9 mm. Limited by this range, we can only test the
reasonability of the first and second stages, as shown in Fig. 4. We introduce how to use the rainfall
and debris flow data recorded in Appendix 1 to calculate $P_{df}$: (1) The original AEP value is rounded
to one decimal place, and the rounded AEP are listed in the 8th column of Appendix 1, which were
sorted from largest to smallest; (2) the maximum $AEP_i$ was set to 90 mm, and $[AEP_i, AEP_i-5]$ was
used as the search window to collect the rainfall events and debris flow events; and (3) we count the
number of debris flow events $N_{df}$ and the number of rainfall events $N_{rain}$ in each search window and
then calculate $P_{df}=N_{df}/N_{rain}$. Based on the above steps, the collected data and calculated $P_{df}$ are listed
in Table 2. As shown in Table 2, a positive correlation between the probability of debris flow
occurrence and AEP in JJG was determined. When AEP < 10 mm, a total of 205 rainfall processes
were recorded; however, no debris flow events were observed, and the debris flow occurrence
probability was 0, which is consistent with the results of Stage 1 derived from Dens-ID.


Table 2 Collected and calculated $P_{df}$ in each search window

|  | Field observation data and calculated $P_{df}$ | | | | | | | | | | |
|---|---|---|---|---|---|---|---|---|---|---|---|
| AEP intervals | 0-10 | 10-15 | 15-20 | 20-25 | 25-30 | 35-40 | 35-40 | 45-50 | 50-55 | 70-75 | 75-80 |
| $N_{df}$ | 0 | 3 | 2 | 7 | 7 | 4 | 4 | 5 | 3 | 1 | 1 |
| $N_{rain}$ | 205 | 133 | 111 | 127 | 124 | 106 | 106 | 49 | 31 | 8 | 5 |
| $P_{df}$ (%) | 0 | 2.3 | 1.8 | 5.5 | 5.6 | 3.8 | 3.8 | 10.2 | 9.7 | 12.5 | 20.0 |

Based on $P_{df}$ and AEP listed in Table 2, their relationship can be described by the exponential

function denoted as $P_{df} = 1.5917e^{0.031AEP}$, which is similar to that of Stage 2 in Fig.4. Therefore,
two $P_{df}$-AEP curves derived from field observation data and the Dens-ID model were obtained for
further analysis, as shown in Fig.5. The two curves were nearly parallel. Eq. 12 is used to analyze
the correlation of the two curves, and $r$ is equal to 0.93, suggesting they have a very high correlation.
Therefore, the function of $P_{df} = f(AEP)$ derived from Dens-ID, which is used to describe the
evolution trend of debris flow occurrence probability with AEP variation, is reasonable.

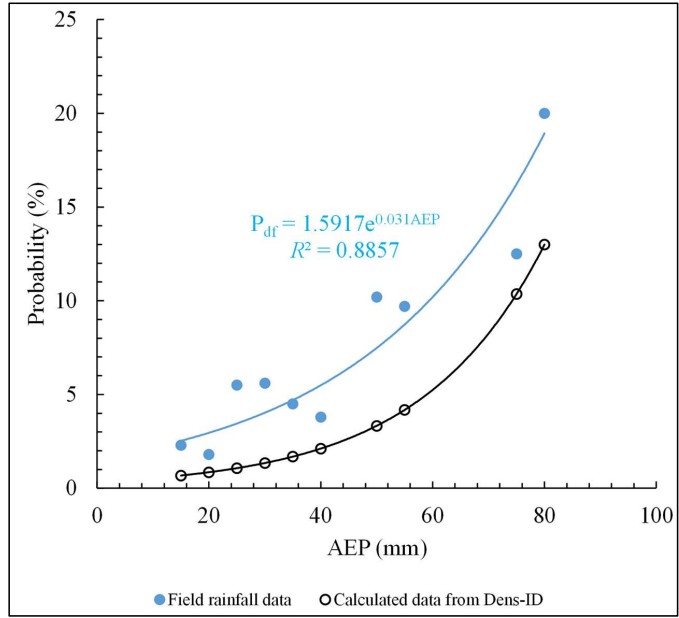


Fig.5 Relationship of AEP and $P_{df}$ obtained from field observation data and Dens-ID model (the blue line is



derived from field observation data, and the black line is derived from Dens-ID)

We can also see from Fig.5 that although the variation tendencies of the two curves are

consistent, there is a significant bias between them. As shown in Fig.5, the blue line fitted through

the observation data is above the black line derived from Dens-ID, indicating that Dens-ID

underestimated the probability of debris flow occurrence if the observation data were used as the

reference. However, we cannot conclude that there is a precision problem in the calculation results

of the Dens-ID. (1) Although 1094 rainfall processes and 37 debris flow events are field observation

data, there are many uncertain factors in Eq. 7 for calculating AEP using these rainfall data (Kim et

al., 2021), such as the subjectivity existing in $K$ and $n$ of Eq. 7, which render uncertainty in the

calculated AEP. In this case, if the data in Appendix 1 are used as the real value for evaluating the

precision of Dens-ID, the error evaluation result may be unfair to Dens-ID. In this case, it is unfair

to evaluate the Dens-ID error by using the calculated AEP in Appendix 1 as the true value. However,

this uncertainty can show consistent directional deviations because of the fixed values of $K$ and $n$ in

Eq.7; therefore, the uncertainty has no effect on the correlation analysis. (2) To establish the

functional relationship between $P_{df}$-AEP, a large number of rainfall scenarios were simulated using

the Dens-ID model. Dens-ID simulated 3376, 3182, 2677, and 2677 rainfall processes with AEP =

20, 40, 45, and 50 mm, respectively. The total number of simulated rainfall processes was

significantly larger than that of the 1094 observed rainfall events. The collected 1094 rainfall events

still cannot fully reflect all rainfall conditions in nature; that is, the amount of the observed 1094

rainfall data is still inadequate when used as the denominator for calculating the probability of debris

flow occurrence in JJG. Therefore, the $P_{df}$ calculated using the field observation data may be

generally higher than that calculated using Dens-ID. (3) Dens-ID cannot fully and accurately



describe the formation process of the debris flow in JJG because of the simplification in theory and
boundaries. Dens-ID is also affected by the accuracy of the input parameters (Zhang et al., 2020),
which may eventually lead to deviations between the simulation results and field observations.
**5 Conclusions**
The Dens-ID model was used to derive the ID threshold curves corresponding to different AEP
in the JJG. Thus, the Monte Carlo integral equation was used to construct the function of $P_{df}$-AEP
for a probability density distribution of field observation rainfall data. The functional relationship
was verified using a large amount of field observation data from JJG. The following conclusions
were drawn.
The qualitative conclusion recognized by scholars that "the greater the AEP, the higher the
probability of subsequent rainfall triggering debris flow" is described by a clear mathematical
equation in this study. For the probability of debris flow occurrence in JJG, the effective range of
AEP that can affect debris flow formation was verified as 10–110 mm. Based on the simulation
results, the probability of debris flow occurrence in JJG is 0 when AEP < 10 mm, and the relationship
between $P_{df}$ and AEP can be described by an exponential function when 10 mm≤AEP≤85 mm.
Limited by the total volume of provenance supply, infiltration capacity of topsoil, and soil saturation,
when the AEP is greater than 85 mm, the growth rate of the probability curve slows, and the
maximum $P_{df}$ stabilizes at 18.96%. The plausibility of the first two evolution stages of the $P_{df}$-AEP
piecewise function is effectively confirmed by the field observation data because the $P_{df}$-AEP
relationship obtained from field observation data is highly correlated with the simulation results of
Dens-ID. However, the reasonability of the last two stages of the $P_{df}$-AEP piecewise function cannot
be tested because of the lack of field observation data, and the errors of the $P_{df}$-AEP piecewise



450 function cannot be verified because of the uncertainty of the AEP derived from the observation

451 rainfall data.

452   This study mathematically confirms that "the greater the AEP, the higher the probability of

453 subsequent rainfall triggering debris flow" and quantifies this qualitative conclusion using piecewise

454 functions. This can effectively reveal the essential relationship between the two natural events of

455 rainfall and debris flow, quantitatively describe the impact of different AEPs on the probability of

456 debris flow occurrence, and provide key technical support for the early warning of debris flows.

457 **Data availability**

458 No data sets were used in this article.

459 **Author contributions**

460 All the authors searched, collected, and process the historical rainfall and debris flow data from JJG

461 field observation station. Shaojie Zhang and Hongjuan Yang wrote parts of the paper, Both Shaojie

462 Zhang and Kaiheng Hu are the Supervision and Funding acquisition, Juan Ma is responsible for

463 Formal analysis, and Dunlong Liu is responsible for data curation.

464 **Competing interests**

465 The contact author has declared that none of the authors has any competing interests.

466 **Acknowledgement**:

467 This work was supported by the Chongqing Municipal Bureau of Land, Resources and Housing

468 Administration (KJ-2021016), Project of the Department of Science and Technology of Sichuan

469 Province (No. 2021YFG0258), National Natural Science Foundation of China (No. 42001100).

470 **References**

471 Abraham, M.T., Satyan, N., Rosi, A., Pradhan, B., Segoni, S.: Usage of antecedent soil moisture for





improving the performance of rainfall thresholds for landslide early warning. Catena, 200,

105147, 2021.

Abraham, M.T., Satyam, N., Pradhan, B., Alamri, A.M.: Forecasting of landslides using rainfall
severity and soil wetness: A probabilistic approach for Darjeeling Himalayas. Water (Switzerland)
12, 1–19, 2020.
Adams, B., Fraser, H., Howard, C., Hanafy, M.: Meteorological data analysis for drainage system
design. J. Environ. Eng. 112, 1986.
Albert, G.E.: A general theory of stochastic estimates of the Neumann series for solution of certain
Fredholm integral equations and related series, in: M.A. Meyer (Ed.), Symposium of Monte Carlo
Methods, Wiley, New York, 1956.
Bel, C., Liébault, F., Navratil O., Eckert N., Bellot H., Fontaine, F., Laigle, D.: Rainfall control of
debris-flow triggering in the Réal Torrent, Southern French Prealphs, 291, 17-32, 2017.
Bennett, G.L., Molnar, P., Mcardell, B.W., Burlando, P.: A probabilistic sediment cascade model of
sediment transfer in the Illgraben. Water Resources Research, 50, 1225-1244, 2014.
Bernard, M., Gregoretti, C.: The use of rain gauge measurements and radar data for the model-based
prediction of runoff-generated debris flow occurrence in early warning systems. Water Resources
Research, 57, e2020WR027893, 2021.
Berti, M., Simoni, A.: Experimental evidences and numerical modelling of debris flow initiated by
channel runoff. Landslides, 3, 171-182, 2005.
Calvo, B., Savi, F.: A real-world application of Monte Carlo procedure for debris flow risk
assessment, Computers and Geosciences, 35(5), 967-977, 2009.
Castillo, V.M., Gómez-Plaza, A., Martínez-Mena, M.: The role of antecedent soil water content in



the runoff response of semiarid catchments: a simulation approach. Journal of Hydrology, 284,

114-130, 2003.

Chen, C.W., Oguchi, T., Chen, H., Lin, G.W. Estimation of the antecedent rainfall period for mass
movements in Taiwan, Environmental Earth Sciences, 77, 184, 2018.
Chen, C.W., Saito, H., Oguchi, T.: Analyzing rainfall-induced mass movements in Taiwan using the
soil water index, Landslides, 14, 1031-1041, 2017.
Coe, J.A., Kinner, D.A., Godt, J.W. Initiation conditions for debris flows generated by runoff at
Chalk Cliffs, central Colorado. Geomorphology, 3, 270-297, 2008.
Crozier, M.J.: Landslides: causes, consequences & environment. Croom Helm, London, p 25, 1986.
Cui, P., Zhu, Y.Y., Chen, J., Han, Y.S., Liu, H.J.: Relationships between antecedent rainfall and
debris flows in Jiangjia Ravine, China. In: Chen & Major, eds., Debris-Flow Hazards Mitigation:
Mechanics, Prediction, and Assessment, Millpress, Netherlands, 3-10, 2007.
De Vita, P.: Fenomeni d'instabilita` delle coperture piroclastiche dei Monti Lattari, di Sarno e di
Salerno (Campania) ed analisi degli eventi pluviometrici determinanti. Quad. Geol. Appl., 7, 213–

239, 2000.

Donovan, I.P., Santi, P.M. A probabilistic approach to post-wildfire debris-flow volume modeling,
Landslides, 14(4): 1345-1360, 2017.
Fiorillo, F., Wilson, R.C. Rainfall induced debris flows in pyroclastic deposits, Campania (southern
Italy). Engineering Geology, 75, 263-289, 2004.
Gabet, E.J., Mudd, S.M.: The mobilization of debris flows from shallow landslides. Geomorphology

1, 207-218, 2006.

Han, Z., Chen, G.Q., Li, Y.G., He, Y.: Assessing entrainment of bed material in a debris-flow event:



a theoretical approach incorporating Monte Carlo method: Assessing Entrainment of Bed
Material by Debris Flow, Earth surface processes and landforms, 40(14): 1877-1890, 2015.
Hirschberg, J., Badoux, A., McArdell, B.W., Leonarduzzi, E., Molnar, P.: Evaluating methods for
debris-flow prediction based on rainfall in an Alpine catchment. Nat. Hazards Earth Syst. Sci.,

21, 2773-2789, 2021.

Hong, M., Kim, J., Jeong, S.: Rainfall intensity-duration thresholds for landslide prediction in South
Korea by considering the effects of antecedent rainfall. Landslides, 15, 523–534, 2018.
Hu, W., Xu, Q., Wang, G.H., van Asch, T.W.J., Hicher, P.Y.: Sensitivity of the initiation of debris
flow to initial soil moisture. Landslides 12, 1139–1145, 2015.
Huang, C.H.: Critical rainfall for typhoon-induced debris flows in the Western Foothills, Taiwan.
Geomorphology, 185, 87-95, 2013.
Hürlimann, M. Coviello, V., Bel, C., Guo, X.J., Berti, M., Graf, C., Hübl, J., Miyata, S., Smith, J.B.,
Yin, H.Y.: Debris-flow monitoring and warning, Review and examples. Earth-Science Reviews,

199, 102981, 2019.

Iverson, R. M., Reid, M. E., LaHusen, R. G.: Debris Flow Mobilization from Landslides. Annu. Rev.
Earth Planet, 25: 85-138, 1997.
Jones, R., Thomas, R.E., Peakall, J., Manville, V.: Rainfall-runoff properties of tephra: Simulated
effects of grain-size and antecedent rainfall. Geomorphology, 282, 39-51, 2017.
Kim, S.W., Chun, K.W., Kim, M., Catani, F., Choi, B., Seo, J.: Effect of antecedent rainfall
conditions and their variations on shallow landslide-triggering rainfall thresholds in South Korea.
Landslides, 18, 569-582, 2021.
Kohler, M.A., Linsley, R.K.: Predicting the runoff from Storm Rainfall. US Department of



Commerce, Weather Bureau, Washington, D.C, 1951.
Le Bissonnais, Y., Renaux, B., Delouche, H. Interactions between soil properties and moisture
content in crust formation, runoff and interrill erosion from tilled loess soils. Catena, 25(1), 33-

46, 1995.

Li, L., Zhang, S.X., Li, S.H., Qiang, Y., Zheng, Z., Zhao, D.S.: Debris Flow Risk Assessment
Method Based on Combination Weight of Probability Analysis, Advances in civil engineering,

2021, 1-12, 2021.

Liu, D.L., Zhang, S.J., Yang, H.J., Zhao, L.Q., Jiang, Y.H., Tang, D., Leng, X.P.: Application and
analysis of debris-flow early warning system in Wenchuan earthquake-affected area. Nat. Hazards
Earth Syst. Sci., 16, 483-496, 2016.
Liu, X.L., Wang, F., Nawnit, K., Lv, X.F., Wang, S.J. Experimental study on debris flow initiation.
Bulletin of Engineering Geology and the Environment, 79, 1565-1580, 2020.
Long, K., Zhang, S.J., Wei, F.Q., Hu, K.H., Zhang, Q., Luo, Y. A hydrology-process based method
for correlating debris flow density to rainfall parameter and its application on debris flow
prediction. Journal of Hydrology, 589, 125124, 2020.
Luk, S.H.: Effect of antecedent soil moisture content on rainwash erosion. Catena, 12, 129-139,

1985.

Marra, F., Destro, E., Nikolopoulos, E.I., Zoccatelli, D., Creutin, J.D., Guzzetti, F., Borga, M.:
Impact of rainfall spatial aggregation on the identification of debris flow occurrence thresholds.
Hydrol. Earth Syst. Sci., 21, 4525-4532, 2017.
Paola, F.De., Risi, R.De., Crescenzo, G. Di, Giugni, M., Santo, A., Speranza, G.: Probabilistic
Assessment of Debris Flow Peak Discharge by Monte Carlo Simulation, Journal of Risk and



Uncertainty in Engineering Systems, Part A: Civil Engineering, 3(1), A4015002, 2017.
Papa, M.N., Medina, V., Ciervo, F., Bateman, A.: Derivation of critical rainfall thresholds for

shallow landslides as a tool for debris flow early warning systems. Hydrol. Earth Syst. Sci. 17,

4095-4107, 2013.

Peres, D.J., Cancelliere, A.: Derivation and evaluation of landslide-triggering thresholds by a Monte

Carlo approach. Hydrol. Earth Syst. Sci., 18, 4913-4931, 2014.

Peres, D.J., Cancelliere, A.: Modeling impacts of climate change on return period of landslide

triggering. Journal of Hydrology, 567, 420-434, 2018.

Richards, L.A. Capillary condition of liquids in porous mediums. Physics 1, 318–333, 1931.
Schoener, G., Stone, M.C.: Monitoring soil moisture at the catchment scale-A novel approach

combing antecedent precipitation index and rader-derived rainfall data, Journal of Hydrology,

589, 125155, 2020.

Segoni, S., Piciullo, L., Gariano, S.L.: A review of the recent literature on rainfall thresholds for

landslide occurrence, Landslides, 15:1483-1501, 2018b.

Segoni, S., Rosi, A., Lagomarsino, D., Fanti, R., Casagli, N.: Brief communication: Using averaged

soil moisture estimates to improve the performances of a regionalscale landslide early warning

system. Nat. Hazards Earth Syst. Sci. 18, 807–812, 2018a.

Senthilkumar, V., Chandrasekaran, S.S., Maji, V.B.: Geotechnical characterization and analysis of

rainfall-induced 2009 landslide at Marappalam area of Nilgiris district, Tamil Nadu state, India.

Landslides, 14, 1803-1814, 2017.

Tang, H., Mcguire, L.A., Kean, J.W., Smith, J.B.: The impact of sediment supply on the initiation

and magnitude of runoff-generated debris flows. Geophysical Research Letters, 47,





e2020GL087643, 2020.
Thomas, M.A., Collins, B.D., Mirus, B.B.: Assessing the feasibility of satellite-based thresholds for
hydrologically driven landsliding. Water Resource Research, 55, 9006-9023, 2019.
Tisdall, A.: Antecedent soil moisture and its relation to infiltration. Aust. J. Agric. Res., 2 (3), 342–

348, 1951.

Van Genuchten, M.: A closed form equation for predicting the hydraulic conductivity of unsaturated
soils. Soil Sci. Soc. Am. J. 44, 892–898, 1980.
Wei, F.Q., Hu, K.H., Zhang, J., Jiang, Y.H., Chen, J.: Determination of effective antecedent rainfall
for debris flow forecast based on soil moisture content observation in Jiangjia Gully, China. In:
DeWrachien, D., Brebbia, C.A., Lenzi, M.A., eds., Monitoring, Simulation, Prevention and
Remediation of dense debris flows II. WIT Transactions on Engineering Sciences, England. 13-

22, 2008.

Yan, Z.Z., Hong, Z.M.: Using the Monte Carlo method to solve integral equations using a modified
control variate. Applied mathematics and computation, 242,764-777, 2014.
Yang, H.J., Zhang, S.J., Hu, K.H., Wei, F.Q., Wang, K., Liu S.: Field observation of debris flow
activities in the initiation area of Jiangjia Gully, Yunnan Province, China, Journal of Mountain
Science, 19(6): 1602-1617, 2022.
Zhang, S.J., Xu, C.X., Wei, F.Q., Hu, K.H., Xu, H., Zhao, L.Q., Zhang, G.P.: A physics-based model
to derive rainfall intensity-duration threshold for debris flow. Geomorphology, 351, 106930, 2020.
Zhang, S.J., Yang, H.J., Wei, F.Q., Jiang, Y.H., Liu, D.L.: A model of debris flow forecast based on
the water-soil coupling mechanism. Journal of Mountain Science, 25, 757-763, 2014.
Zhao, B.R., Dai, Q., Han, D.W., Dai, H.C., Mao, J.Q., Zhuo, L.: Probabilistic thresholds for





landslides warning by integrating soil moisture conditions with rainfall thresholds, Journal of
Hydrology, 574, 276-287, 2019a.
Zhao, B.R., Dai, Q., Han, D., Dai, H., Mao, J., Zhuo, L., Rong, G.: Estimation of soil moisture using
modified antecedent precipitation index with application in landslide predictions. Landslides 16,
2381–2393, 2019b.
Zhu, Y.J., Shao, M.G.: Variability and pattern of surface moisture on a small-scale hillslope in
Liudaogou catchment on the northern Loess Plateau of China, Geoderma, 147, 185-191, 2008.
Zhuang, J.Q., Cui, P., Wang, G.H., Chen, X.Q., Iqbal, J., Guo, X.J.: Rainfall thresholds for the
occurrence of debris flows in Jiangjia Gully, Yunnan Province, China. Eng. Geol. 195, 335–346,

2015.