# Peer review of "Investigation of the functional relationship between antecedent rainfall and the probability"

_Hydrology and Earth System Sciences, 2022_

## Author Comment (AC2)

**Response to Reviewer2**

The authors appreciate the reviewer to carefully review our manuscript. The reviewer has given some very professional suggestions. The authors believe that the professional suggestions will be very helpful to improve the readability of our manuscript.

According to the author's understanding, the presented suggestions by the reviewer are summarized into the following three points: (1) the necessary of classifying the stages into the stage 3, stage 4; (2) more detailed explanation on the stages in Fig.4 is needed; (3) more works are necessary for assessing the relation between Pdf and AEP.

All of the suggestions are so important for the authors, we will totally accept the three comments and try our best to modify our manuscript.

---

## Author Response (AR1)

**Modification explanation for reviewers**

We are very grateful to the reviewers for their affirmation of the manuscript "Investigation of the functional relationship between antecedent rainfall and the probability of debris flow occurrence in Jiangjia Gully, China" (*Number*: *hess-2022-263*)", which is of great significance to authors. What moved us even more was that the reviewers were so busy that they could give such detailed comments here. From these comments, the author feels the rigorous scholarship, knowledge and responsibility of the reviewers. Thanks again!

According to the Reviewers' comments, we have amended this manuscript by using the track changes mode in MS Word.

**Response to Reviewer 1**:

**(1) Reviewer 1**: Jiangjiagou is ideal for DF study because the extensive monitoring data are available and preliminary studies. However, it is still necessary to provide more environmental settings, such as climate, geomorphometry, vegetation, and soil types, since they all highly related to DF generation and behavior.

**Authors**: The authors have provided detailed information including environmental settings, such as climate, geomorphometry, vegetation, and soil types. Please check lines 98-109 of Pages 5 in the tracking version.

**(2) Reviewer 1**:it is worthy to substantially improve the language. I have trouble to follow this work due to its language. BUT I found it becomes easier for me if it is literally, word by word, translated into Chinese, which is my first language.

**Authors**: The authors have asked the native English speaker to improve languages of the manuscript in order to improve the readability of this manuscript. All the polishing contents are saved in the tracking mode.

**(3) Reviewer 1**: The description of the Dens-ID model also needs to be improved. Terms used in

equations are lost.

**Authors**: The authors have carefully checked each equation relating to Dens-ID, and we have mended some mistakes and terms missing in equations. Please check lines 149, 157-158. We also tired our best to improve the description of the Dens-ID model, Please check lines 127-199.

**(4) Reviewer 1**: Results can also be presented in a better and more concise way. Discussion should be separated from results for this study. I'd like to read a comprehensive discussion on insights of certain results and observations. For instance, we already know the small DF probability compared with rainfall from existing data. What is the additional information the simulations provide? What can I learn by reading the simulated data? Is the Pdf and AEP curve unique to Jiangjiagou or how can I transfer this curve to other watersheds? Could you compare the different stages of this curve with previously studies of DF in this watershed and other watersheds in different geographic environments?

**Authors**: We have modified the results and Discussions have been separated from results. We followed the reviewer's suggestions and tried our best to improve the Discussions contents. Please check lines 492-543.

**(5) Reviewer 1**: Additionally, I am wondering how the Den-ID model represent the DF generation without consideration of momentum law. It seems the occurrence of a DF is determined by the density of the soil-water mixture, in which soil and water are estimated by a safety factor equation and a infiltration method, respectively.

**Authors**: Thank you very much for raising this issue, Dens-ID model does not consider the momentum method. In fact, the model only uses conservation of mass to calculate the total amount of unstable solid matter and the total amount of runoff in the whole basin, and assumes that these sources can flow to the location of the basin outlet. The model then calculates the density of soil and water mixture at the basin outlet through Eq.6.

**Response to Reviewer 2**:

**(1) Reviewer 2**: Fig.4, Is it necessary to classify the stages into the stage 3, stage 4? Somehow, the explanation of the stages is vague. The reason of this classification is not clear.

**Authors**: The authors have carefully considered this very important suggestion. Just like the issue proposed by the reviewer, we also find that explanation of the stages 3 is vague. Therefore, we decided to delete Stage 3 and Stage 4. Then we found power function is the best fitting cure. The detailed modifications are listed in Lines 401-432 in the tracking version.

**(2) Reviewer 2**: Fig.4, Moreover, it seems that the stage 3 has too little data to analyze the result. In the stage 4, it has stable single Pdf value against AEP, contrary to the general knowledge of the debris flow occurrence. Interpretation of these matters has to be described. Anyway, more explanation on the stages is needed.

**Authors**: The authors have followed the first suggestion and deleted the stage. Please check lines 401-432

**(3) Reviewer 2**: Fig.5, (1) This relationship is should be validated using AUC (ROC) analysis or the critical index such as Threat Score.

**Authors**: In Fig.5, the two curves are fitted through field observation data and Dens-ID model. The two curves are nearly parallel. Eq. 12 was used to analyze the correlation of the two curves, and $r$ is equal to 0.93, suggesting they have a very high correlation. However, a significant bias is existed between them. Basically, the probability value derived from the field observation data is larger than that from the Dens-ID model in the condition of a given AEP. Accordingly, the authors cannot use AUC (ROC) analysis or the critical index to verify the Eq.14. We analyze the error of equation 14 using field observation data as reference, and find that the error is significantly high. The authors try the best to draw several reasons inducing the situation of high error. The most important point is insufficient field observation data causing the small amount of rainfall events and a higher $P_{df}$ comparing to the $P_{df}$ from Dens-ID. With the accumulation of rainfall observation data of JJG, it is believed that the $P_{df}$ derived from field observation data will gradually decrease until it is close to the calculated result of Dens-ID model. The above contents are listed in the manuscript. Please check lines 459-468. We also discussed the reason causing the large bias, Please check lines 535-543.

**(4) Reviewer 2**: The authors intension on this matter is not obvious (How to evaluate the achievement of this research?)

**Authors**: We claimed the intension on this research in Discussion 5.2. Please check them in lines 491-543.

---

## Author Response (AR2)

**Modification explanation for Editors**

We are very grateful to the editors for the professional and detailed comments on the manuscript "Investigation of the functional relationship between antecedent rainfall and the probability of debris flow occurrence in Jiangjia Gully, China" (*Number*: *hess-2022-263*)". According to the Editors' comments, we have amended this manuscript by using the track changes mode in MS Word.

**(1) Editor**: Line 162: The word Equation should be written in full.

**Authors**: The word Equation has been written in full, please check line 147 in the new edited tracking version. Additionally, the authors found that there are many others should be mended according to this suggestion. All of the "Eq." are rewritten in full. For example, please check lines 130, 135, 139, 146 etc. in the new edited tracking version.

**(2) Editor**: Line 170: Can you please check the derivation of d_s from equation 3? This is not clear since this equation does not depend on d_s.

**Authors**: Thank you for your careful reviewing on the manuscript, the editor is right. The authors have deleted the sentence of "Using $d_s$ derived from Eq. 3 as input" in order to avoid to mislead the editor and other potential readers.  Please check line 156 in the new edited tracking version.

**(3) Editor**: Line 171. I guess the word should be unstable instead of "instable"

**Authors**: The authors have changed "instable" to "unstable", please check line 157 in the new edited tracking version.

**(4) Editor**: Line 182: Please check wording in this sentence.

**Authors**: The authors carefully read this sentence and also think the wording should be modified. The sentences are now amended as "after firstly presetting the density of the water-soil mixture as $\rho_{mix}$, Dens-ID also needs to simulate many rainfall scenarios including long durations with low-intensity rainfall and short durations with high-intensity rainfall in order to obtain a sufficient number of $[D_i, l_i]$". Please check lines 166-168 in the new edited tracking version.

**(5) Editor**: Line237: Can you please provide the EAP values range and explain they increase by an interval of size 5?

**Authors**: The authors have added some necessary sentences to further explain. The detailed contents are as follows: Since AEP in JJG ranges in 0-88 mm according to the measured rainfall data, Dens-ID presets several AEP values including 10, 15, 20, 25, 30, 35, 40, 45, 50, 55, 60, 65, 70, 75, 80, 85. The cases of AEP=0 and AEP=5 mm are excluded, because the two cases represents such a low initial rainfall condition that any ID curve cannot derived from Dens-ID. The purpose of increasing AEP by an interval of size 5 is to get adequate ID curves, which will be helpful to calculate $P_{df}$ under different AEP conditions. Please check lines 217-222 in the new edited tracking version.

**(6) Editor**: Line 371: Can you improve formatting of equation 14?

**Authors**: The authors have improved the format of Equation 14 and added several sentences to further introduce this fitted equation. Please check lines 342-345 in the new edited tracking version.

**(7) Editor**: Line 460: Can you please check the wording in this line?."is existed" should be replaced by existed.

**Authors**: The authors have modified this word. Please check line 409 in the new edited tracking version.

**(8) Editor**: It is unusual to include additional authors at a later review stage of a manuscript. We would appreciate the author's description about the contributions of the newly added authors, and how their contribution help improving your manuscript. A statement about all author's contribution during the manuscript preparation and during the review process is very much appreciated.

**Authors**: The authors are very appreciated. We have prepared another file to state the contributions of all the authors during the manuscript preparation and during the review process. We also listed the contribution of each author in the manuscript as follows:

**During the preparation process:**

Shaojie Zhang: Writing original draft, Conceptualization, Funding acquisition

Hongjuan Yang: Data curation

Kaiheng Hu: Supervision

Juan Ma: Formal analysis

During the review process, the works of language editing and modifying this manuscript needs the help of Xiaohu Lei and Fangqiang Wei. Their contributions to this manuscript are listed as follows:

Xiaohu Lei: Translate the new Chinese sentences in the modified version into English, and be responsible for modifying the images, such as Figure 4 in the text.

Fangqiang Wei: Give the more professional suggestions on how to modify this manuscript according to the reviewers' comments, and be responsible for Writing - Review & Editing before and after the works of polishing languages.